# Association between Friends’ Use of Nicotine and Cannabis and Intake of Both Substances among Adolescents

**DOI:** 10.3390/ijerph18020695

**Published:** 2021-01-15

**Authors:** Rachel Herold, Rachel Boykan, Allison Eliscu, Héctor E. Alcalá, Maciej L. Goniewicz

**Affiliations:** 1Department of Pediatrics, T 11-040, Renaissance School of Medicine at Stony Brook University, Stony Brook, NY 11794, USA; rachel.herold@stonybrookmedicine.edu (R.H.); Allison.Eliscu@stonybrookmedicine.edu (A.E.); 2Department of Family, Population and Preventive Medicine, Renaissance School of Medicine at Stony Brook University, Stony Brook, NY 11794, USA; hector.alcala@stonybrookmedicine.edu; 3Department of Health Behavior, Division of Cancer Prevention and Population Sciences, Roswell Park Comprehensive Cancer Center, Elm and Carlton Streets, Buffalo, NY 14263, USA; maciej.goniewicz@roswellpark.org

**Keywords:** tobacco, cannabis, marijuana, peer group, adolescent, biomarkers

## Abstract

Nicotine and cannabis use are common among adolescents and may be associated with behavioral problems, poor academic outcomes and use disorders. The goals of this analysis were the following: (1) Describe the influence of friends’ nicotine and cannabis smoking and vaping on self-reported use. (2) Describe the relationship between friends’ nicotine and cannabis use on participants’ urinary biomarkers of nicotine (cotinine) and cannabis (11-nor-9-carboxy-Δ⁹tetrahydrocannabinol=THC-COOH). This is a secondary analysis of survey and biomarker data collected in adolescents aged 12–21 between April 2017 and April 2018, in Long Island, New York. Bivariate and multivariable analyses were conducted using SPSS 26. A cutoff value of ≥10 ng/mL was used to signify recent usage for urinary cotinine and THC-COOH levels. Over one-third of the 517 surveyed adolescents reported using tobacco and one-third reported using cannabis. A significant relationship between friends’ substance use and self-use was found. For both tobacco and cannabis, over 90% (*p* < 0.01) of participants with urinary biomarker levels above cutoff had friends who used the respective substance. Friends’ nicotine and friends’ cannabis use were each independently associated with urinary biomarker levels for those substances (for nicotine, beta = 88.29, *p* = 0.03; for cannabis, beta = 163.58, *p* = 0.03). Friends’ use of nicotine and cannabis is associated with adolescents’ intake, as well as the physiological exposure to those substances. These findings underscore the importance of including peer influence in the discussion with adolescents about tobacco and cannabis use.

## 1. Introduction

Nicotine and cannabinoids, two of the most commonly used substances by adolescents, are primarily delivered by inhalation either from smoked products or vaping devices (e-cigarettes). In 2020, almost 20% of high-school students were current (past-month) e-cigarette users; most reported vaping nicotine-containing products [1]. In 2019, 22% of high-school seniors were current cannabis users, and 14% had vaped cannabis in the past month [2]. While the long-term effects of vaping nicotine are still unclear, adolescent use of nicotine is associated with behavioral problems, depression, anxiety, and an increased risk of nicotine addiction [3]. Adolescents who smoke cannabis are similarly at risk for significant morbidity, including poor academic outcomes and cannabis-use disorder [4,5,6]. Vaping cannabis may exacerbate this risk, as cannabis formulations used in vaping devices may contain highly concentrated (14–28%) tetrahydrocannabinol (THC) [4,7,8].

While the influence of peers on adolescent nicotine and cannabis use is well described [9,10,11], this association has not been examined using biomarkers. Biomarkers have been used to quantify and confirm self-report. Benowitz et al. found a significant correlation between nicotine and cannabinoid usage patterns and urinary biomarkers (cotinine=nicotine metabolite and THC-COOH=THC metabolite) [12]. We previously reported that adolescent self-reported nicotine and cannabinoid use corresponded to urinary cotinine and THC-COOH levels, and that the co-use of tobacco and cannabis was common and reflected in elevated biomarker levels. Specifically, we found that 55% of past-week e-cigarette users also used cannabis, and that 67% of past-week tobacco smokers and 67% of dual users of combusted tobacco and e-cigarettes used cannabis [13]. We aimed to further examine nicotine and cannabis usage in participants from our previously published study by increasing understanding of the socio-environmental impact on biomarker levels. The goals of this analysis were as follows: (1) Describe the influence of friends’ nicotine and cannabis smoking and vaping on self-reported nicotine and cannabis use. (2) Describe the relationship between friends’ nicotine and cannabis use on participants’ urinary biomarkers of nicotine (cotinine) and cannabis (THC-COOH).

## 2. Materials and Methods

This study is a secondary analysis of survey and biomarker data collected in adolescents, ages 12–21, between April 2017 and April 2018, previously described in detail [13]. Analyses were conducted using SPSS 26. Nicotine and cannabis users were categorized into 5 groups: daily, weekly, monthly, ever and never users. Nicotine use included use of nicotine-containing e-cigarettes or tobacco cigarettes. Cannabis use included any form of cannabinoid (Table 1). The previously established cutoff (≥10 ng/mL) [13], indicating recent (past-week) usage, was applied to urinary cotinine and THC-COOH levels for bivariate analyses (i.e., Pearson chi-square). For multivariable analyses (i.e., multivariable linear regression), continuous measures of cotinine and THC-COOH were used.

## 3. Results

As Table 2 shows, for this analysis, all 517 participants’ surveys were analyzed; a total of 265 had biomarker data. Among all 517 surveyed, over one-third (197/517) reported ever using nicotine (38%); one in three participants (31%) reported ever using cannabis. Of ever nicotine users, 37% reported smoking tobacco and the vast majority reported vaping e-cigarettes (95%). The majority (91%) of ever cannabis users smoked and 36% vaped cannabis. Among ever e-cigarette nicotine users, 51% were past-month users and 23% were past-week users. Among ever cannabis users (any form), 54% were past-month and 36% were past-week users.

Common reasons for initial nicotine use were curiosity (65% of tobacco smokers; 69% of e-cigarette users) and friends’ use (39% of tobacco smokers; 51% of e-cigarette users). Friends’ use was positively associated with self-use of nicotine: 75% (χ^2^_1_ = 117.36, *p* < 0.01) of ever nicotine users and 89% (χ^2^_1_ = 4.26, *p* < 0.04) of past-month nicotine users reported having friends using nicotine (either smoked or vaped). Although curiosity was the most commonly reported reason for initial cannabis use (72%), 41% reported friends’ use as a reason for first trying cannabis. Chi-square testing revealed a statistically significant relationship between having friends who used cannabis and one’s own cannabis use, both ever (71%, χ^2^_1_ = 175.00, *p* < 0.01) and past-month (82%, χ^2^_1_ = 11.41, *p* < 0.01).

Almost all (93%) of participants with cotinine levels above cutoff had friends who used nicotine (χ^2^_1_ = 21.87, *p* < 0.01). Multivariable linear regression revealed that friends’ use of nicotine was associated with cotinine concentration, when controlling for own nicotine use (Beta = 88.29, *p* = 0.03). Similarly, 90% of participants with THC-COOH above cutoff had friends who used cannabis (χ^2^_1_ = 28.13, *p* < 0.01). Multivariable linear regression revealed that friends’ use of cannabis (Beta = 163.58, *p* = 0.03) was associated with THC level, when controlling for own cannabis use.

## 4. Discussion

In this analysis, friends’ nicotine and cannabis use were each independently associated with adolescents’ self-reported usage of each substance, and with urinary biomarkers. One explanation for this finding may be that friends’ nicotine and cannabis use reinforces one’s own use behaviors, perhaps leading to more frequent or heavier use of each substance. While we did not quantify time spent with friends or assess attitudes regarding friends’ influence, our prior analysis of e-cigarette users in this study group demonstrated higher biomarker levels with more frequent use [13].

It is also possible that the independent effect of friends’ nicotine and cannabis use reflects secondhand exposure. Although exposure to secondhand tobacco smoke is associated with the presence of urinary cotinine in adolescents [14], the effect of secondhand nicotine exposure from aerosol (vaping) is less well studied. Similarly, research has shown that measurable levels of urinary THC-COOH can be found after passive exposure to marijuana smoke [15]. With the increasing potency of cannabis products available today and the pervasive use of these substances among adolescents, this potential effect may have more significant implications.

Regardless of the explanation, our findings underscore the important influence of friends’ behavior in adolescents’ nicotine and cannabis use, and the potential physiologic impact of this behavior, particularly in light of the known morbidity associated with the increasingly high concentrations of nicotine and cannabinoids in products used today.

The cross-sectional design of this study limits drawing conclusions beyond association. Additionally, as cannabis users reported using multiple forms of cannabis, we were unable to differentiate the potential influence of different cannabis formulations (e.g., smoked vs. vaped) on friends’ use and on biomarker levels. Further study should evaluate whether product type and/or strength contribute to these findings.

## 5. Conclusions

We found that friends’ use of nicotine and cannabis were independently associated with adolescents’ use of each substance, and this was reflected in urinary biomarker levels. These findings underscore the important influence of friends’ behavior in adolescent substance use and physiologic impact. Pediatricians should discuss friends’ influence on behavior and physiology when counseling their adolescent patients on the smoking and vaping of both nicotine and cannabis products.

## Figures and Tables

**Table 1 ijerph-18-00695-t001:** Questions asked of participants.

Have you ever used an electronic vape product, even one or two times?
Why did you try an electronic vape product the first time?
When was the last time you used an electronic vape product?
Please think about the electronic vape product you most recently used. Does it have nicotine?
Do your friends use electronic vape products when you are with them?
Have you ever tried cigarette smoking, even one or two puffs?
Why did you try cigarettes the first time?
When was the last time you smoked a cigarette?
Do your friends smoke cigarettes when you are with them?
Have you ever used marijuana in any form, even one or two times?
Why did you try marijuana the first time?
When you use (or used) marijuana does (or did) the product also contain nicotine?
When was the last time you used any type of marijuana?
Do your friends use marijuana when you are with them?

**Table 2 ijerph-18-00695-t002:** Sample Characteristics (*n* = 517).

	*n*	Frequency (%)
Ever used nicotine	197	38
Smoked tobacco (among nicotine users)	72	37
Vaped e-cigarettes (among nicotine users)	Ever: 187	Ever: 95
	Past month: 95	Past month: 51
	Past week: 43	Past week: 23
Used cannabis any form	Ever: 162	Ever: 31
	Past month: 87	Past month: 54
	Past week: 58	Past week: 36
Smoked cannabis (among cannabis users)	147	91
Vaped cannabis (among cannabis users)	58	36

## Data Availability

The data presented in this study are available on request from the corresponding author. The data are not publicly available due to confidentiality of information reported by study participants.

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
