# Peer review of "Association between Friends’ Use of Nicotine and Cannabis and Intake of both Substances among Adolescents"

_ijerph, 2021, doi:10.3390/ijerph18020695_

Round 1

Reviewer 1 Report

The report is well written and clearly describes the objective. It is very interesting, and infrequent, the use of biomarkers to obtain evidence of validity of the self-report responses of the participants in a survey. The reasons why young people use cannabis and tobacco are also very interesting, but some issues should be clarified.

In the Materials and Methods section, it is indicated that bivariate and multivariate analyses have been carried out, but the bivariate analyses are not specified. I deduce that they are correlations looking at the results, but it should be indicated in this section, as they do with multivariate analyses indicating the performance of multiple linear regressions. The authors also do not explain why the cut-off point for nicotine. In this sense, if a cut-off point is established, I do not understand that correlations have been made (if this has been the case), since in that case the variables should be continuous. Perhaps the cut-off point has been established afterwards to find out how many young people consume above it. This must be clarified.

In the Results section it is indicated that 517 participants responded to the survey, but only 265 of them had biomarkers. Have the percentages described below been calculated from the total survey participants? If so, it must be indicated. On the other hand, the use of biomarkers to validate the responses of the participants in the survey is not clearly seen. The authors should clarify.

Finally, the bibliographic references must be inserted in the text before the punctuation marks. For example, in line 40 you write "month. [2]", when you should write "month [2]." It is convenient to review all the text.

Author Response

Reviewer 1:

The report is well written and clearly describes the objective. It is very interesting, and infrequent, the use of biomarkers to obtain evidence of validity of the self-report responses of the participants in a survey. The reasons why young people use cannabis and tobacco are also very interesting, but some issues should be clarified.

In the Materials and Methods section, it is indicated that bivariate and multivariate analyses have been carried out, but the bivariate analyses are not specified. I deduce that they are correlations looking at the results, but it should be indicated in this section, as they do with multivariate analyses indicating the performance of multiple linear regressions. The authors also do not explain why the cut-off point for nicotine. In this sense, if a cut-off point is established, I do not understand that correlations have been made (if this has been the case), since in that case the variables should be continuous. Perhaps the cut-off point has been established afterwards to find out how many young people consume above it. This must be clarified.

In the Results section it is indicated that 517 participants responded to the survey, but only 265 of them had biomarkers. Have the percentages described below been calculated from the total survey participants? If so, it must be indicated. On the other hand, the use of biomarkers to validate the responses of the participants in the survey is not clearly seen. The authors should clarify.

Survey responses are reported for all 517 surveyed. This has been clarified in the beginning of the results section, lines 75-76, and by the addition of Table 1.  The use of biomarkers to validate responses was reported in our first paper, referenced in this secondary report (reference #13).

Finally, the bibliographic references must be inserted in the text before the punctuation marks. For example, in line 40 you write "month. [2]", when you should write "month [2]." It is convenient to review all the text.

Thank you for noticing these details. This has been corrected throughout the manuscript.

Reviewer 2 Report

Review of: Association between friends’ use of nicotine and cannabis and intake of both substances
among adolenscents.
Mansuscript number: ijerph-1062906
Summary: This is a brief report looking at how friends influence nicotine and cannabis smoking and vaping and attempts to find a relationship between friends urinary biomarkers of nicotine and cannabis.

They used secondary data from a survey from Long Island New York for adolescents between 12 and 21. They make the claim that those who test positive for these substances have friends who also use cannabis.

Overall Comments: The manuscript is well written and is well referenced. The methodology portion is too vague to determine exactly what analyses were performed. Just to say multiple regression analysis was used does not tell the reader what variables were in the multiple regressions, which has a huge impact on the associated p-values of all variables in the model. Good Science requires one to be able to reproduce the work. And from a secondary analysis you should expect the reader to be able to obtain the data and arrive at the same conclusions. In order to arrive at the same conclusions the exact
analyses used need to be presented. Just showing a p-value isn’t sufficient.

The other item not addressed is the pervasive use of cannabis and nicotine. Only very sheltered children would not have a friend or an acquaintance who used either compound.

Specific Comments:
1. Don’t require the reader to go back and read the survey from another paper. Put the relevant questions here is this paper.
2. When you say you used Bivariate analyses. Which ones? Simple Linear Regression? Simple Logistic Regression? Correlation Analysis? Categorical Data Analysis?
3. For your Multiple Linear Regression analysis you report Beta. However, there are many betas in a multiple linear regression analysis. I would rather you perform a standardized multiple linear regression analysis as the beta can be interpreted in terms of standard deviation changes in the means.
4. There is no mention of what analysis was used for friends use and self use. 75% (p<0.01) is reported but I have no idea how I could reproduce this number from the data since there is no method attached to it.

5. Were there any hypotheses that you tested that didn’t come up significant? Many times good research can be found where there are hypotheses that didn’t come up significant but one would think they should or that they were close to the cut-off. It is worth stating any p-values that are below 0.1 as they may indicate that further study should be given to this hypothesis.

6.  However, any evidence towards associating nicotine and cannabis use should be presented.  They need to fix up the analysis section because virtually no details are provided on how they arrived at their conclusions.

Author Response

Reviewer 2:

Review of: Association between friends’ use of nicotine and cannabis and intake of both substances
among adolenscents.
Mansuscript number: ijerph-1062906
Summary: This is a brief report looking at how friends influence nicotine and cannabis smoking and vaping and attempts to find a relationship between friends urinary biomarkers of nicotine and cannabis.

They used secondary data from a survey from Long Island New York for adolescents between 12 and 21. They make the claim that those who test positive for these substances have friends who also use cannabis.

Overall Comments: The manuscript is well written and is well referenced. The methodology portion is too vague to determine exactly what analyses were performed. Just to say multiple regression analysis was used does not tell the reader what variables were in the multiple regressions, which has a huge impact on the associated p-values of all variables in the model. Good Science requires one to be able to reproduce the work. And from a secondary analysis you should expect the reader to be able to obtain the data and arrive at the same conclusions. In order to arrive at the same conclusions the exact
analyses used need to be presented. Just showing a p-value isn’t sufficient.

Please see response to Reviewer # 1, above. Additionally, chi square and degrees of freedom (df) are now provided  - see lines 85-96.

The other item not addressed is the pervasive use of cannabis and nicotine. Only very sheltered children would not have a friend or an acquaintance who used either compound.

Specific Comments:
1. Don’t require the reader to go back and read the survey from another paper. Put the relevant questions here is this paper.

The questions are now included as Table 1.

  1. When you say you used Bivariate analyses. Which ones? Simple Linear Regression? Simple Logistic Regression? Correlation Analysis? Categorical Data Analysis?

Please see responses to other reviewer, above. This has now been specified in the edited paper. Specifically please see lines 71-73, 89, 93, 96.

  1. For your Multiple Linear Regression analysis you report Beta. However, there are many betas in a multiple linear regression analysis. I would rather you perform a standardized multiple linear regression analysis as the beta can be interpreted in terms of standard deviation changes in the means.
  2. Were there any hypotheses that you tested that didn’t come up significant? Many times good research can be found where there are hypotheses that didn’t come up significant but one would think they should or that they were close to the cut-off. It is worth stating any p-values that are below 0.1 as they may indicate that further study should be given to this hypothesis.
  3. However, any evidence towards associating nicotine and cannabis use should be presented.  They need to fix up the analysis section because virtually no details are provided on how they arrived at their conclusions.

Co-use was addressed in our prior work (ref. # 13). We have added previously reported co-use rates in this draft, for a reference point, adding the following line in the introduction, in lines 55-57:

“Specifically, we found that 55% of past-week e-cigarette users also used cannabis, and 67% of past-week tobacco smokers and 67% of dual users of combusted tobacco and e-cigarettes, used cannabis [13].”

Additional details of analysis have been added in response to other reviewer questions.

Reviewer 3 Report

The present study is an interesting article to the readers to realize the effect of friends’ use of some substances on the adolescents. However, it is difficult to read the results without a clear table. Please make one table to enroll all the uni-variate analysis ( at least two major independent variables (curiosity and friends use) that could be revealed a crude OR and its 95% CI additionally) and multi-variable analysis, either nicotine using as the dependent variable or cannabis using as the dependent variable.

Author Response

Reviewer 3:

The present study is an interesting article to the readers to realize the effect of friends’ use of some substances on the adolescents. However, it is difficult to read the results without a clear table. Please make one table to enroll all the uni-variate analysis ( at least two major independent variables (curiosity and friends use) that could be revealed a crude OR and its 95% CI additionally) and multi-variable analysis, either nicotine using as the dependent variable or cannabis using as the dependent variable.

We have added a table (table 2) to better illustrate the characteristics of participants in this subanalysis. The point estimates are now included within the text, with clarification of the statistics in the methods, as requested by reviewers 1 and 2. We felt this approach would clarify matters.However, if you still believe we should add another table we can do so. Thank you.